

# NCBP2 predicts the prognosis and the immunotherapy response of cancers: a pan-cancer analysis

Shichao Li[1,*], Yulan Wang[1,*], Xi Yang[2], Miao Li[3], Guoxiang Li[2], Qiangqiang Song[1] and Junyu Liu[4]

[1] Department of Pathology, General Hospital of Xinjiang Military Command, Urumqi, Xinjiang, China
[2] Department of Medical Service, General Hospital of Xinjiang Military Command, Urumqi, Xinjiang, China
[3] School of Rehabilitation Medicine, Xinjiang Medical University, Urumqi, Xinjiang, China
[4] Department of Gastroenterology, The Third Affiliated Hospital of Sun Yat-Sen University, Guangzhou, Guangdong, China
[*] These authors contributed equally to this work.

Corresponding authors
Shichao Li, lishichao1208@126.com
Junyu Liu, ljysci@foxmail.com

## ABSTRACT

**Background**. The cap-binding complex (CBC) plays a crucial role in facilitating gene expression by safeguarding mRNA from nonsense-mediated decay, promoting mRNA splicing, 3′-end processing, and facilitating nuclear export. Nevertheless, the precise biological functions and clinical implications of CBC in cancer remain ambiguous, necessitating further investigation for clarification.

**Methods**. The present study utilized the cBioPortal database to investigate the genetic alterations of nuclear cap binding protein subunit 2 (NCBP2) in pan-cancer. The Cancer Genome Atlas (TCGA) and online web tools were employed to analyze the correlation between NCBP2 and prognosis, genome instability, immune infiltration, immune response, cancer stemness, and chemotherapeutic efficacy in pan-cancer. Furthermore, the expression of NCBP2 was confirmed by immunohistochemistry (IHC) and functional analysis at the single-cell level was conducted using the CancerSEA database.

**Results**. *NCBP2* exhibited distinct genetic alterations in pan-cancer with an increased expression in 24/32, while decreased expression in 3/32, types of cancers. IHC confirmed the aberrant expression of NCBP2 in lung squamous cell carcinoma (LUSC), pancreatic adenocarcinoma (PAAD), kidney renal papillary cell carcinoma (KIRP) and kidney renal clear cell carcinoma (KIRC). NCBP2 was correlated with overall survival (OS), disease-specific survival (DSS), and progression-free survival (PFS) in various cancers. Importantly, it was identified as a risk factor for OS, DSS and PFS in PAAD and uterine corpus endometrial carcinoma (UCEC). Gene Set Enrichment Analysis (GSEA) demonstrated that elevated NCBP2 was linked to immune and proliferation related pathways across multiple cancer types. Furthermore, a negative association between NCBP2 and stromal score, immune score, and ESTIMATE score was detected, and a positive correlation was observed between NCBP2 and diverse immune cells as well as stemness-indexes in the majority of cancer types. Drug sensitivity analysis revealed that drugs associated with NCBP2 primarily targeted DNA replication, chromatin histone methylation, ABL signaling, cell cycle, and PI3K signaling. Additionally, an examination at the single-cell level indicated that NCBP2 was positively correlated with cell cycle progression, DNA damage, DNA repair, invasion, and stemness in most cancer
types, while negatively correlated with apoptosis, inflammation, and hypoxia in certain cancers.

**Conclusion**. In this study, we revealed the correlation of NCBP2 with prognosis, microenvironment and stemness, indicating that NCBP2 might be a potential therapeutic target for more effective and personalized therapy strategies in pan-cancer.

## INTRODUCTION

Globally, cancer stands as a prominent contributor to mortality rates and carries substantial implications for the economy and society (*Soerjomataram & Bray, 2021*). Tumorigenesis encompasses various mechanisms, such as cancer cell proliferation and survival, tumor microenvironment, and tumor immunity (*Siegel et al., 2021*). Patients diagnosed with cancer at advanced stages generally experience a more unfavorable prognosis compared to those diagnosed at an early stage. Despite significant advancements in surgical techniques, chemotherapy targeted therapy and immunotherapy, cancer remains a formidable obstacle, as patients encounter reduced prognoses and survival rates attributed to factors including drug resistance and adverse effects (*Siegel et al., 2021*). Consequently, the identification of novel biomarkers and treatment targets for cancer assumes paramount importance in enhancing human well-being.

Two different mechanisms are involved in cap-dependent translation of mRNAs: eukaryotic initiation factor 4E(eIF4E)-dependent translation (ET) and cap-binding complex (CBC)-dependent translation (CT) (*Gonatopoulos-Pournatzis & Cowling, 2013*). The m7G cap-binding protein eIF4E can lead to tumor initiation and progression (*Avdulov et al., 2004*), and is a promising target for cancer therapies (*Boussemart et al., 2014*). Moreover, targeting ET regulates tumor microenvironment and sensitizes tumors to anti-PD-1 immunotherapy (*Guo et al., 2021*; *Wang et al., 2022*). These studies emphasize the importance of cap-binding proteins in cancers. The CBC is also essential for efficient gene expression, and it protects mRNA from nonsense-mediated decay and promotes mRNA splicing, 3′-end processing and nuclear export (*Topisirovic et al., 2011*). CBC components have shown prognostic potential in oral squamous cell carcinoma (*Arora et al., 2023*) and promoted pancreatic cancer progression (*Xie et al., 2023*). However, the role of CBC in modulating the tumor immune microenvironment is still largely unknown, and need to be further clarified.

Cancer stem cells (CSCs) refer to a small proportion of tumor cells that possess self-renewal and differentiation ability (*Batlle & Clevers, 2017*). The existence of CSCs has been validated in non-solid (*Bonnet & Dick, 1997*; *Lapidot et al., 1994*) and solid tumors (*O'Brien et al., 2006*). Stemness is defined as the ability of self-renewal and differentiation (*Saygin et al., 2019*). Higher stemness, and the more likely it is to have distant metastasis and multitherapy resistance, ultimately leading to cancer progression and poor prognosis (*Fabregat, Malfettone & Soukupova, 2016*; *Prasetyanti & Medema, 2017*; *Walcher et al.,*

*2020*; *Zhang et al., 2022*). In previous study, a one-class logistic regression (OCLR) machine-learning algorithm was employed to extract transcriptomic and epigenetic features from non-transformed pluripotent stem cells and their differentiated progeny, and generate stemness indices which identifies dedifferentiated oncogenic state (*Malta et al., 2018*). These stemness indices could be used to evaluate the degree of tumor dedifferentiation and survival prognosis of patients (*Prasad et al., 2020*).

Considering the role of cap-binding proteins in cancers and its potential role in cancer immunity, the CBC genomic alteration pattern of pan-cancer was revealed and we choose the gene *NCBP2*, which had the highest alteration frequency, for further investigation. We designed the pan-cancer analysis to investigate the expression, prognostic value, functional enrichment, immunotherapy predictive ability, stemness and potential targeting drugs of NCBP2 in various cancers. Collectively, this study provides evidence for elucidating immunotherapeutic effect of NCBP2 in a variety of cancers, which is likely to provide useful information for further research.

## METHODS

### Data source and processing

Transcriptional data and clinical information from the TCGA Pan-cancer cohort, as well as normal human tissue data from the Genotype-tissue expression (GTEx) database, were acquired through university of Cingiforria Sisha Cruz (UCSC) Xena (*Goldman et al., 2020*) (https://xenabrowser.net/). The expression of NCBP2 in TCGA and GTEx datasets was analyzed with transcripts per million (TPM) using the same sequencing platform and library preparation to reduce potential batch effects. The abbreviations for different types of cancers can be found in Table S1.

Two immunotherapy studies were included in this study. The sequencing and clinical data of 298 patients with urological cancer who received atezolizumab treatment (IMvigor210 cohort) were obtained from the website research-pub.gene.com/IMvigor210CoreBiologies/packageVersions/. The cohort of melanoma patients treated with nivolumab (anti-PD1) (GSE91061) was acquired from the Gene Expression Omnibus database (GEO).

The TIMER (*Li et al., 2016*) 2.0 database (http://timer.cistrome.org/) was utilized to acquire NCBP2-associated immune cell infiltration correlations for the TCGA Pan-Cancer study. We summarized the correlations between NCBP2 mRNA expression and 21 immune cell subsets, including CD4+ T cells, CD8+ T cells, $\gamma\delta$T cells, T cell regulatory (Tregs), T cell follicular helper (Tfh), cancer-associated fibroblast (CAF), endothelial cells (Endo), eosinophils (Eos), mast cells, progenitors of lymphoid/myeloid/monocyte, hematopoietic stem cell (HSC), myeloid-derived suppressor cell (MDSC), B cells, macrophages, dendritic cells, NK cells, nature killer T cell (NKT), monocytes and neutrophils, which were summarized across various cancers using the R package "ggplot2".

Genomic alterations, such as mutations, structural variations, amplifications, deep deletions, and multiple alterations, were obtained from the cBioPortal database (*Gao et al., 2013*; https://www.cbioportal.org/). Furthermore, the cBioPortal web tool was employed to visualize the rate of genomic alterations.
## NCBP2 protein localization and interaction

The Human Protein Atlas (*Uhlen et al., 2010*) provides an interactive presentation showcasing the protein expression profiles of various human tissues. In order to illustrate the subcellular localization of NCBP2, immunofluorescence staining images of human cancer cell lines (U-251, A-431, U2-OS, MCF7, and PC-3) were utilized (Human Protein Atlas).

## Clinical cancer samples

Several cancers samples, including lung cancer, kidney cancer and pancreatic cancer were collected from patients hospitalized in General Hospital of the Xinjiang Military Command from 2018 to 2022. The Medical Ethics Committee of the General Hospital of the Xinjiang Military Command (Urumqi, China) granted approval for the utilization of clinical excisions (2024RR0313), with written informed consent obtained from patients. The collection and utilization of clinical samples adhered strictly to established guidelines, as sanctioned by the Medical Ethics Committee of the General Hospital of the Xinjiang Military Command.

## Immunohistochemistry

Formalin-fixed paraffin-embedded (FFPE) human cancer tissues, including lung squamous cell carcinoma (LUSC), KIRP, PAAD, and KIRC, were sliced into 3 $\mu$m sections and subsequently deparaffinized, rehydrated, and subjected to antigen retrieval. The sections were then incubated overnight at 4 °C with a primary NCBP2 rabbit polyclonal antibody (A7239; ABclonal, Woburn, MA, USA) and subsequently detected using a secondary antibody (goat anti-rabbit). Color development was achieved by applying DAB (K3486, Dako, Santa Clara, CA, USA), and hematoxylin was used as a counterstain. The immunostained sections were scanned using Motic EasyScan.

## Expression and prognosis analysis

The expression levels of NCBP2 between tumor and normal groups in pan-cancer were compared using the "limma" package of R studio software. Additionally, the prognostic value of NCBP2 in pan-cancer was assessed through the utilization of univariate Cox regression analysis. To investigate the relationship between NCBP2 and prognosis, the OS, progression-free interval (PFI), and DSS were chosen as outcome measures. Kaplan–Meier analysis and univariate Cox regression were employed to evaluate the prognostic relevance of NCBP2 for different cancer prognosis types. The NCBP2 FPKM expression data was utilized for the univariate Cox regression analysis, while the bivariate Kaplan–Meier curve analysis was performed using median cutoffs. Subsequently, the log-rank $P$ value and hazard ratio (HR) were determined using the Kaplan–Meier method, and the results were visually depicted on the Kaplan–Meier curve.

## Differentially expressed genes analysis

According to the expression of NCBP2 mRNA, cancer patients were categorized into high-NCBP2 and low-NCBP2 groups, with the top 30% classified as high-NCBP2 patients and the bottom 30% as low-NCBP2 patients. The transcriptional profile in pan-cancer

between these subgroups was analyzed using the "limma" package of R studio software. Genes with an adjusted *P*-value <0.05 were considered as DEGs.

## Gene set enrichment analyses

The hallmark gene set (h.all.v7.5.1. entrez.gmt) was obtained from MSigDB Database (*Subramanian et al., 2005*) (https://www.gsea-msigdb.org/gsea/msigdb/) and employed to compute the normalized enrichment score (NES) and *P* value for each biological process associated with each cancer type (*Subramanian et al., 2005*). Following this, GSEA was performed using the R packages "clusterProfiler" and "ggplot2".

## Immunotherapy prediction analysis

A Spearman correlation analysis was employed to investigate the associations between NCBP2 and tumor mutational burden (TMB), microsatellite instability (MSI). In order to explore the connection between NCBP2 and the response to immune checkpoint blockade (ICB) therapy, data from two sources: IMvigor210, which consisted of 298 bladder cancer patients treated with atezolizumab (anti-PD-L1), and GSE91061, which included 26 melanoma patients treated with nivolumab (anti-PD-1), were utilized. Patients were categorized into low- and high-NCBP2 groups based on the cut-off value determined by the "ROC" R package. The proportion difference between low- and high-NCBP2 cancer groups was evaluated through Chi-square test.

## Stemness index calculation and analysis

In order to assess tumor stemness, six prediction models were acquired from prior studies, which were based on mRNA expression data or DNA methylation data (*Malta et al., 2018*). Subsequently, a model utilizing the aforementioned equation was employed to compute stem cell indices for TCGA samples. The measurement of stemness involves comparing tumor cells and stem cells through an index that ranges from 0 to 1. To investigate the association between NCBP2 and stemness indices, Spearman correlation analysis was conducted and the outcomes were visualized using the R package "ggplot2".

## Drug sensitivity and NCBP2 expression analysis

The expression of NCBP2 in various cell lines was obtained from the Cancer Cell Line Encyclopedia (*Ghandi et al., 2019*) (CCLE; http://sites.broadinstitute.org/ccle/) database. The AUC value, which indicates cellular sensitivity to drugs of paired cell lines, was acquired from the Genomics of Drug Sensitivity in Cancer (GDSC) (*Yang et al., 2013*) database (https://www.cancerrxgene.org/celllines). Subsequently, Spearman correlation was used to estimate the relationship between drug sensitivity and NCBP2. A positive correlation suggests that cell lines exhibiting high NCBP2 expression were resistant to certain drugs, while a negative correlation suggests that cell lines with high NCBP2 expression were sensitive to specific drugs.

## Single cell sequencing analysis

A single-cell assay was conducted on the CancerSEA (*Yuan et al., 2019*) platform (http://biocc.hrbmu.edu.cn/CancerSEA/) to examine the association between NCBP2 and different functional states of various cancers. A correlation threshold of correlation

≥0.3 and a *p*-value<0.05 were established to determine the significance of the relationship between NCBP2 and cancer functional status. Single-cell expression profiles of NCBP2 were visualized using t-SNE diagrams.

## Statistical analyses

To ascertain statistical significance, the Wilcoxon rank-sum test was employed to compare the expression levels of NCBP2 in tumor and normal tissues. The prognostic significance of NCBP2 expression in each cancer type was evaluated through univariate Cox regression and Kaplan–Meier techniques, specifically log-rank tests. Furthermore, a Spearman correlation analysis was performed to examine the statistical relationship between NCBP2 and multiple factors, including levels of immune cell infiltration, genes involved in immune responses, TMB and TMI. To determine the statistical significance of the proportion of ICI-therapy responders and non-responders between low- and high-NCBP2 subgroups, a chi-square test was employed. A *P* value less than 0.05 was deemed to be statistically significant.

# RESULTS

## The genetic alteration of *NCBP2* in pan-cancer

To investigate the role of nuclear cap binding proteins (NCBPs) in cancers, we analyzed the genomic alteration profiles of NCBPs including *NCBP1, NCBP2, NCBP3, NCBP2AS2* and *NCBP2-AS1* and NCBP2L in pan-cancer (TCGA, Firehose Legacy) using cBioPortal database (https://www.cbioportal.org/). As shown in Fig. 1A, *NCBP2* was found to be the most frequently altered of the six genes, with nearly 7% of alterations occurring in pan-cancer, while the remaining genes showed relatively low mutation rates. In addition, Kaplan–Meier survival analysis revealed that patients with *NCBP2* alteration exhibited a shorter OS and disease-free survival (DFS) compared with those without *NCBP2* alterations (Figs. 1B & 1C). Then, we conducted a cancer type-based genetic alteration analysis of *NCBP2* in the TCGA pan-cancer tumor samples. As shown in Fig. 1D, the general alteration frequency of *NCBP2* was relatively high in the top five cancers, which had an alteration frequency of >20% with ''Amplification'' as the primary type, while the most frequent *NCBP2* alteration was identified among LUSC patients (>40%). As a consequence, we further analyzed the relationship between the expression of NCBP2 and copy numbers in the top five cancers. According to our findings, compared with diploid samples, samples with *NCBP2* amplification/gain/deletion exhibited highest/higher/lower mRNA expression level of NCBP2, respectively (Figs. S1A–S1E), indicating that copy numbers of *NCBP2* appeared to be associated with its mRNA expression.

## The expression of NCBP2 in pan-cancer

We further assessed the expression levels of NCBP2 across cancers using integrated data from TCGA and GTEx databases. NCBP2 was differentially expressed in 27 of 32 kinds of cancers. The expression of NCBP2 was higher in LUSC, esophageal carcinoma (ESCA), ovarian serous cystadenocardinoma (OV), cervical squamous cell carcinoma and endocervical adenocarcinoma (CESC), head and neck squamous cell carcinoma (HNSC), uterine carcinosarcoma (UCS), uterine corpus endometrial carcinoma (UCEC),

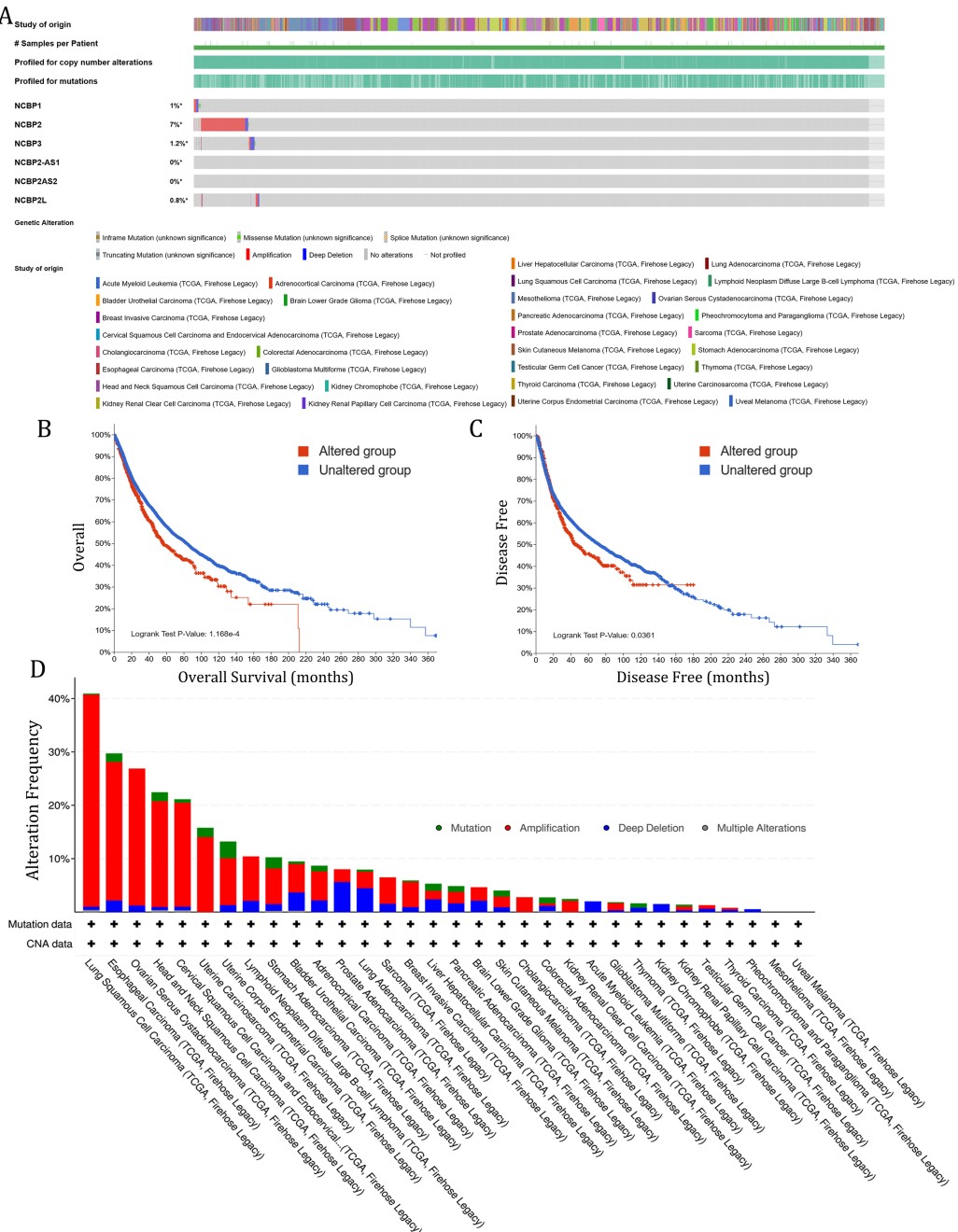

**Figure 1** **The genetic alteration of NCBP2 in pan-cancer.** (A) Genomic alteration profiles of *NCBP1, NCBP2, NCBP3, NCBP2-AS1, NCBP2AS2* and *NCBP2L* in TCGA pan-cancer cohort. (B–C) Kaplan–Meier survival analysis showed OS (B) and DFS (C) in all pan-cancer patients divided by *NCBP2* altered or unaltered groups. (D) The genomic alteration of *NCBP2* in TCGA pan-cancer atlas, including mutation, amplification, deep deletion and multiple alterations.

lymphoid neoplasm diffuse large B-cell lymphoma (DLBC), stomach adenocarcinoma (STAD), bladder urothelial carcinoma (BLCA), lung adenocarcinoma (LUAD), prostate adenocarcinoma (PRAD), breast invasive carcinoma (BRCA), cholangio carcinoma (CHOL), brain lower grade glioma (LGG), pancreatic adenocarcinoma (PAAD), liver hepatocellular carcinoma (LIHC), skin cutaneous melanoma (SKCM), glioblastoma multiforme (GBM), kidney renal papillary cell carcinoma (KIRP), testicular germ cell tumors (TGCT), colorectal carcinoma (COADREAD), thyroid carcinoma (THCA) and thymoma (THYM), compared with corresponding normal tissues. In contrast, the expression of NCBP2 was lower in kidney renal clear cell carcinoma (KIRC), adrenocortical carcinoma (ACC), acute myeloid leukemia (LAML) (expression data for mesothelioma (MESO), pheochromocytoma and paraganglioma (PCPG), sarcoma (SARC) and uveal melanoma (UVM) were not shown for the absence of enough control normal tissue data) (Fig. 2A). As observed in the genetic analysis, most cancers expressed altered levels of NCBP2.

To investigate the location of NCBP2 in cells, the immunofluorescence (IF) staining images from Human Protein Atlas (HPA) database were acquired which indicated that NCBP2 protein was predominantly located in the nucleus and partially in cytoplasm of U-251, A-431, U2-OS, PC3 and MCF7 tumor cell lines (Fig. 2B). Moreover, the expression of NCBP2 protein levels were validated in clinical cancers and paired normal tissues by immunohistochemical staining, which showed that NCBP2 protein were up-regulated in LUSC, KIRP and PAAD compared with normal tissues, since lower expression of NCBP2 was observed in KIRC consistent with previous analysis (Figs. 2C–2F, Figs. S2A–S2D).

## Clinical prognostic significance of NCBP2 in pan-cancer

To further investigate the prognostic potential of NCBP2 in cancers, univariate cox regression and Kaplan–Meier method (log-rank test) were performed for each cancer. A positive relationship between poor OS and NCBP2 was observed in UCEC, PAAD, LIHC, KIRP, KICH and KIRC as shown in the forest plot (Fig. 3A) and patients with high expression of NCBP2 shown worse OS compared to patients with low expression of NCBP2 in all of the 6 cancers (Figs. 3B–3G). Regarding the expression of NCBP2 and DSS, a significant positive association was observed in LIHC, PAAD, LUAD, and UCEC, and a negative association was observed in BLCA (Fig. S3A), and patients (LIHC, PAAD, UCEC) with higher expression of NCBP2 shown worse DSS compared to those with lower expression of NCBP2 (Figs. S3B–S3G). In view of PFI, higher expression of NCBP2 had a favorable influence in BLCA, however, it deemed to be a hazard factor in PAAD and UCEC (Figs. S3H–S3N). It is worth noting that NCBP2 was a risk factor for OS, DSS and PFI of PAAD and UCEC, which were significantly correlated with poor outcomes. Considering that highly expression of NCBP2 was associated with poorer OS for UCEC, PAAD, LIHC, KIRC, KICH and KIRP, multivariate cox regression analysis was performed to further explore the relationship between NCBP2 and OS. As shown in Figs. 3H–3M, NCBP2 was an independent risk factor of OS in PAAD, LIHC and KIRP. Overall, these data suggested that NCBP2 plays a prognostic role in several cancers, but the roles were complicated

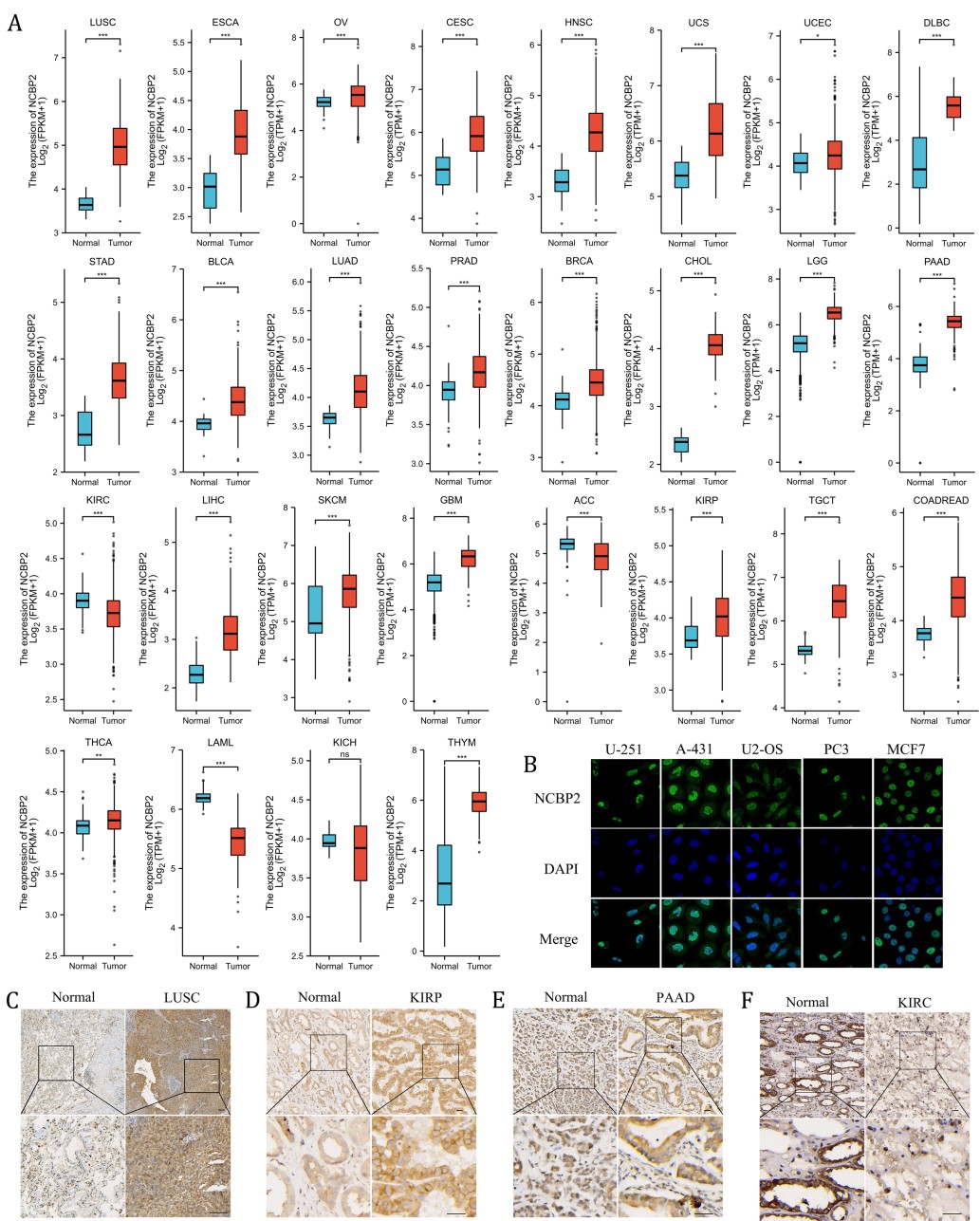

**Figure 2  The expression of NCBP2 in pan-cancer.** (A) The relative mRNA expression level of NCBP2 between tumor and normal tissues in pan-cancer from TCGA cohort. (B) The expression of NCBP2 in several cancer cell lines, including U-251, A-431, U2-OS, PC3 and MCF7 were detected by immunofluorescent staining in Human Protein Atlas. (C–F) The protein expression of NCBP2 in LUSC (C), KIRP (D), PAAD (E) and KIRC (F) and paired normal tissues were detected by immunohistochemical stain. $^*p < 0.05$, $^{**}p < 0.01$, $^{***}p < 0.001$.

and multifaceted across cancers. Further investigation should focus on the function of the NCBP2 protein in cancer cells.

## Gene set enrichment analysis of NCBP2 revealed its association with the cancer immune response

To elucidate the biological processes associated with NCBP2 in cancers, we performed DEGs analysis between the top (71–100%) and bottom (0–30%) NCBP2 expression subgroups in each cancer type. Subsequently, based on the DEGs, GSEA was executed across 33 cancer types to investigate the NCBP2-associated cancer processes. The results showed that immune-related pathways: IFN-alpha response, IFN-gamma response, IL-6/JAK/STAT3 signaling, IL-2/STAT5 signaling, allograft-rejection pathways, inflammatory response, and TNF alpha signaling-via NFKB, were remarkably enriched in a variety of tumors, especially in COAD, HNSC, LUSC and READ (Fig. 4), which indicated that NCBP2 may be involved in tumor immune microenvironment and ligand–receptor interactions between malignant tumor cells and immune cells. Moreover, we also found that NCBP2 was associated with partial proliferative pathways, including G2M checkpoint and E2F targets of many kinds of tumors which possessed significant positive correlation with BRCA, KIRP, LUAD, PAAD, PRAD and STAD, suggesting that NCBP2 could play a significant role in tumor proliferation. In summary, these results provide evidences that the abnormal expression of NCBP2 may be involved in the immune response and progress of cancers.

## Association between immune-related factors and NCBP2

Since NCBP2 was correlated with several immune-related pathways, such as IFN-alpha response and TNF alpha signaling-via NFKB (Fig. 4), we further explored the correlations between NCBP2 and tumor microenvironment through Estimation of STromal and Immune cells in MAlignant Tumor tissues using Expression data (ESTIMATE) algorithm, which was performed to calculate the level of tumor stromal cells, and the infiltration immune cells in tumor tissues based on RNA sequencing data. The correlation between NCBP2 and stromal score, immune score, and ESTIMATE score were shown in Fig. 5A. Notably, NCBP2 expression were negatively associated with stromal score, immune score and ESTIMATE score in almost all cancer types except for ACC, CHOL, DLBC, KICH, LIHC and PAAD.

We further employed the TIMER2.0 database to assess the association between NCBP2 with various tumor infiltrating immune cells, which were determined by a variety of quantitative immune infiltration algorithms based on expression data, to show the infiltration levels of CD4+ T cells, CD8+ T cells, γδT cells, T cell regulatory (Tregs), T cell follicular helper (Tfh), cancer-associated fibroblast (CAF), endothelial cells (Endo), eosinophils (Eos), mast cells, progenitors of lymphoid/myeloid/monocyte, hematopoietic stem cell (HSC), myeloid-derived suppressor cell (MDSC), B cells, macrophages, dendritic cells, NK cells, nature killer T cell (NKT), monocytes and neutrophils in pan-cancer (Fig. 5B). In general, a positive correlation was observed between NCBP2 and various immune cells, such as MDSC, progenitors of lymphoid, CD4+ T cell, CAF, neutrophils and monocytes, however a negative correlation was identified with NKT cells in various

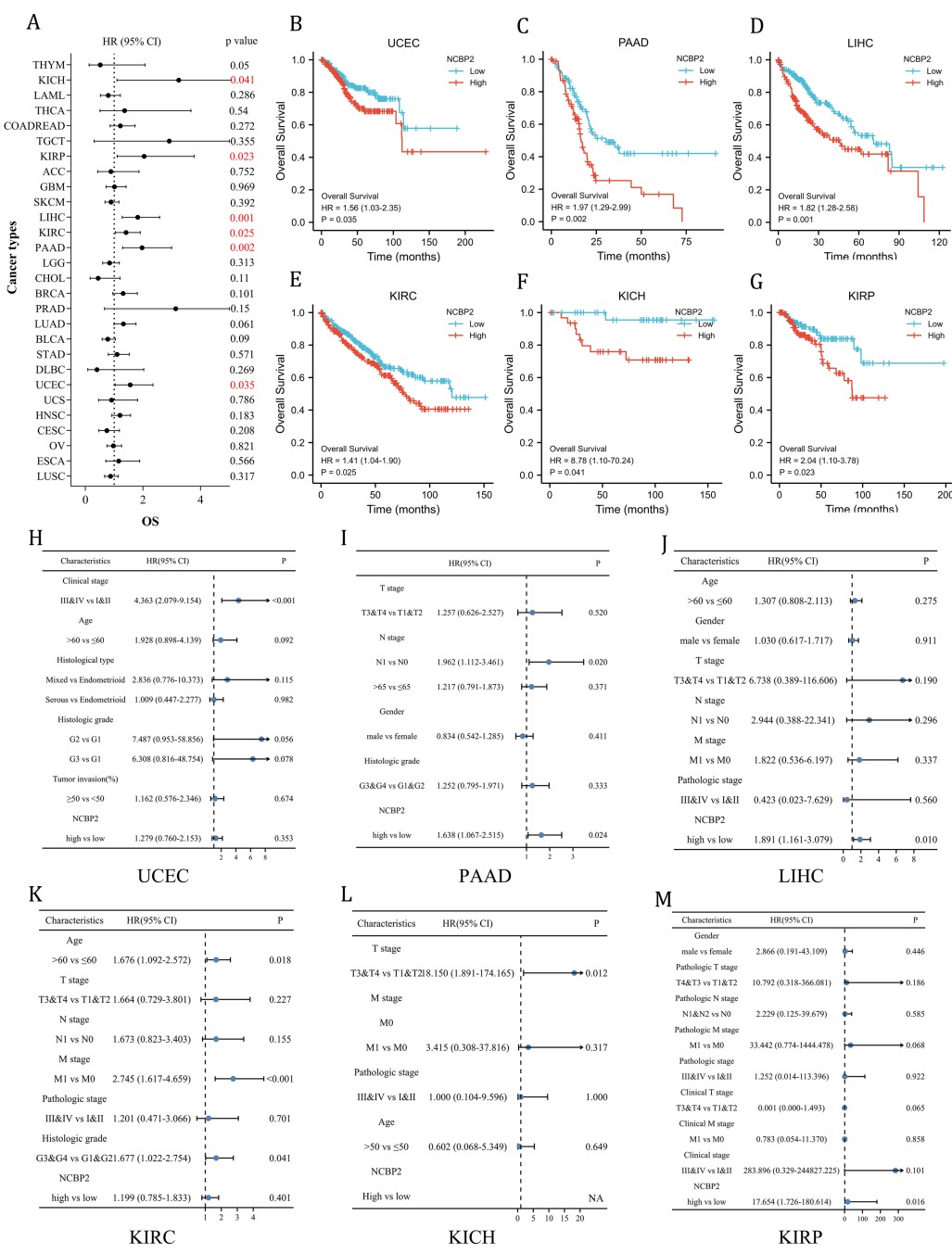

**Figure 3    Clinical prognostic significance of NCBP2 in pan-cancer.** (A) The correlation between NCBP2 and patients' OS were shown by forest plot. (B–G) Kaplan–Meier survival analysis for patients with specific cancers (UCEC, PAAD, LIHC, KIRC, KICH and KIRP) base on the expression of NCBP2. (H–M) Multivariate cox regression analyses evaluated the prognostic independence of NCBP2 and clinicopathological features regarding OS of UCEC, PAAD, LIHC, KIRC, KICH and KIRP in TCGA datasets.

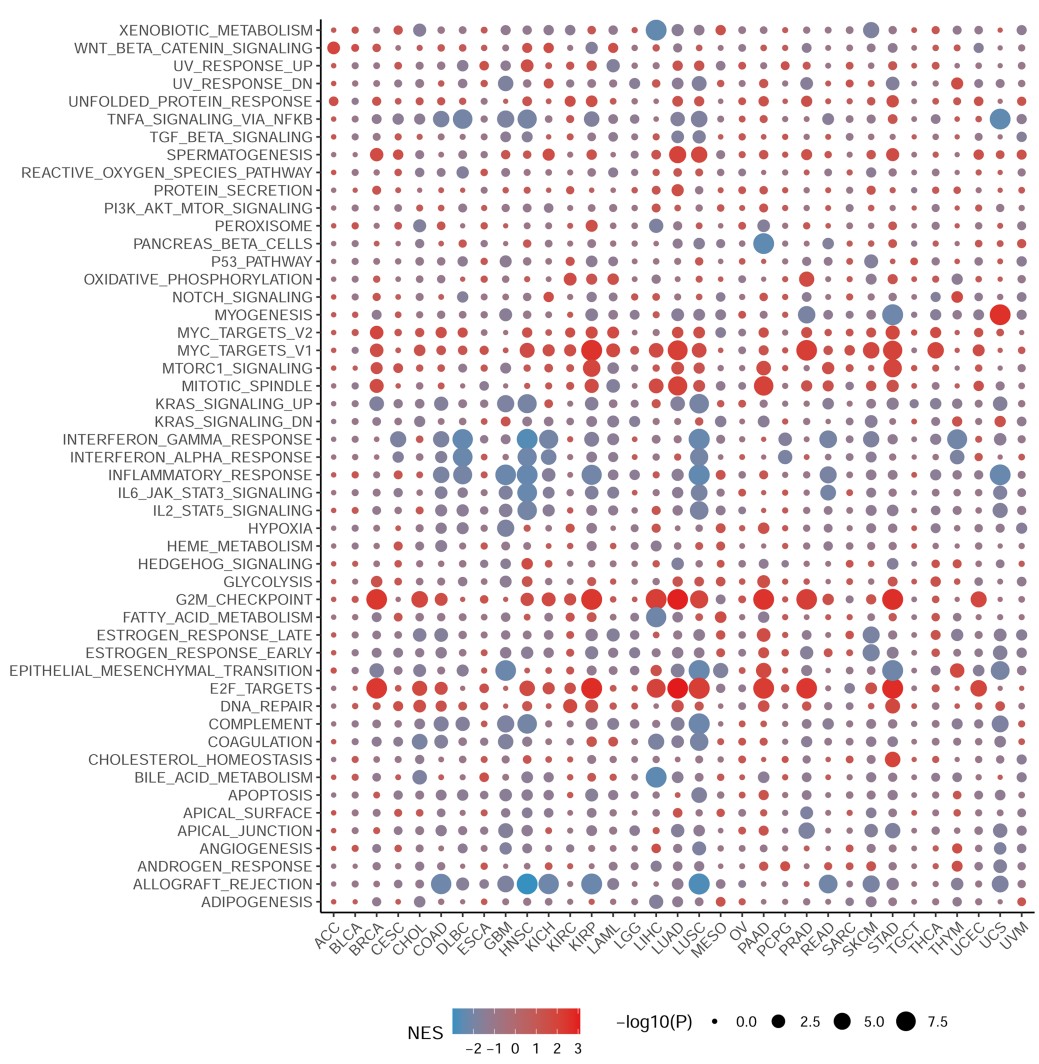

**Figure 4    Gene set enrichment analysis of NCBP2 revealed its association with the cancer immune response.** Differential expressed genes (DEGs) were analyzed between the top and bottom 30% NCBP2 expression subgroup for each cancer in TCGA, following hallmarks gene set enrichment analysis (GSEA) of those DEGs to investigate the NCBP2-associated cancer processes. The size of the circle represents the false discovery rate (FDR) adjusted *P* value of each cancer enrichment item, and the color represents the normalized enrichment score (NES) of each enrichment item.

cancers (Fig. 5B). Likewise, we found that NCBP2 expression was correlated with various immune infiltrating cells in each cancer, and the landscape of this correlation was different, which may be due to the other immune infiltration rates of certain tumors. Interestingly, we found that NCBP2 expression was significantly negatively correlated with CD8+ T cells, dendritic cells and neutrophils in various cancers, including COAD, HNSC and LUSC (Fig. 5B), which exhibited negatively enriched immune-related pathways (Fig. 4).

Due to the essential role of microenvironment in immunotherapy and the potential role of NCBP2 in microenvironment regulation, we aimed to investigate the correlation between NCBP2 and twenty-six immunomodulatory genes using Spearman correlation analysis.

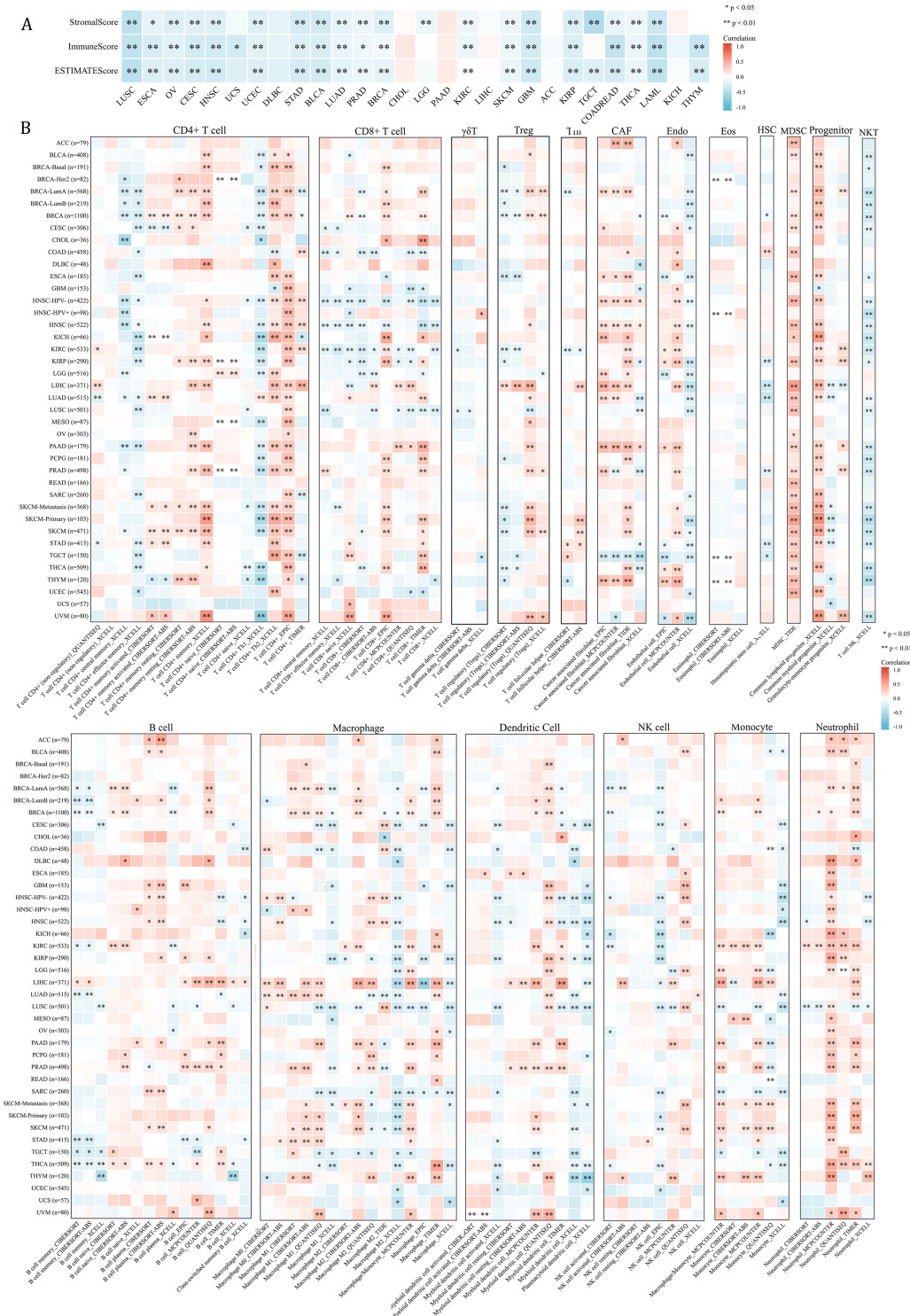

**Figure 5** **Association between tumor microenvironment and NCBP2 expression.** (A) Heatmap showing the correlation of NCBP2 expression and stromal score, immune score, and ESTIMATE score in pancancer. Positive correlation in red and negative correlation in blue. (B) Heatmap showing the correlation of NCBP2 and infiltration levels of immune cells including CD4+ T cells, CAF, progenitor, Endo, Eos, HSC, Tfh, gdT, NKT, regulatory T cells (Tregs), B cells, neutrophils, monocytes, macrophages, dendritic cells, NK cells, Mast cells and CD8+ T cells in cancers. Positive correlation in red and negative correlation in blue. *$p < 1.05$; **$p < 0.01$.

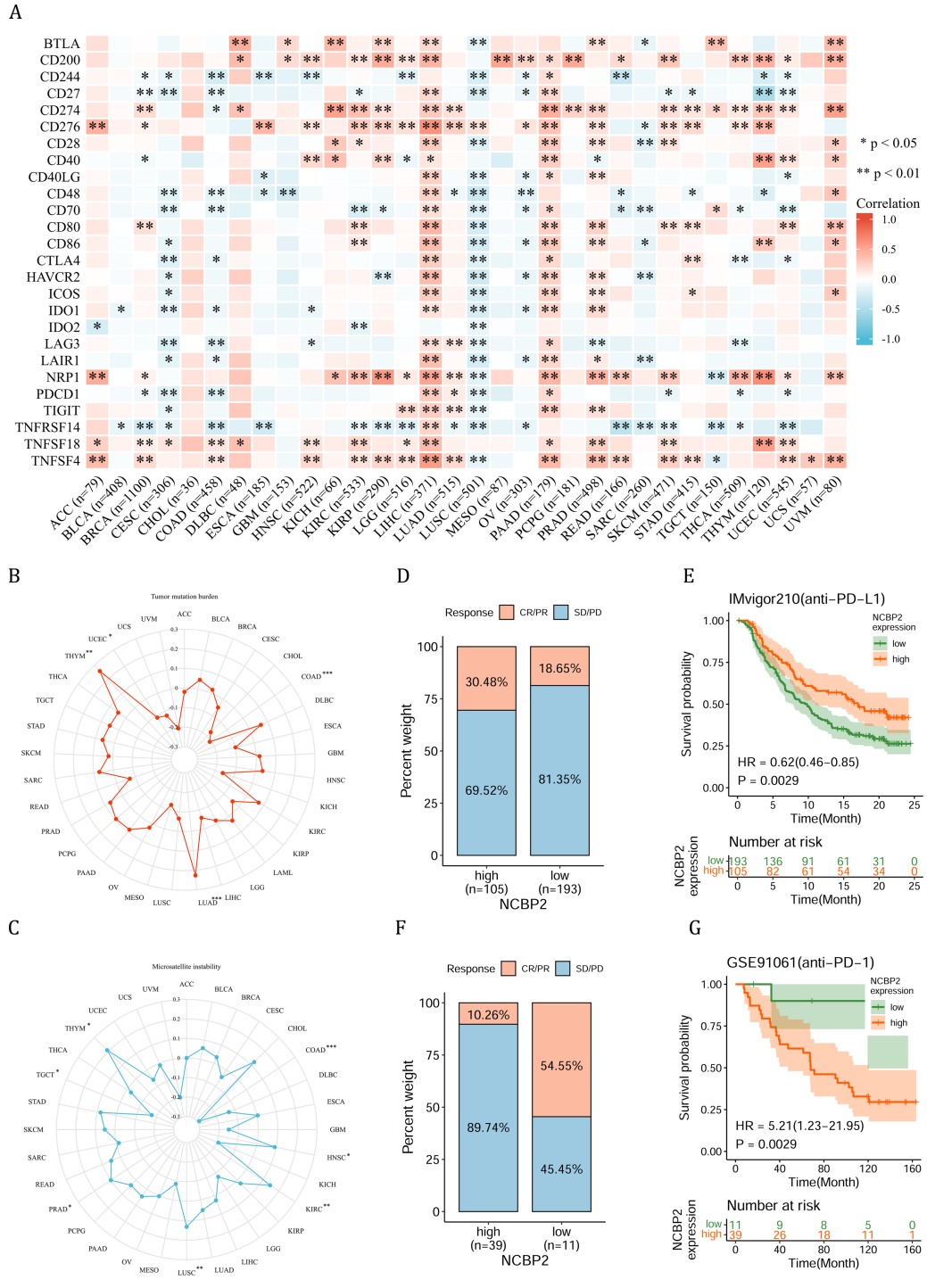

**Figure 6  Association between immune-related factors and NCBP2.** (A) The Spearman correlation heatmap shows the correlation between the expression of NCBP2 and twenty-six immunomodulatory genes in pan-cancer. Red represents positive correlation and blue represents negative correlation. (B–C) Correlations between NCBP2 and tumor mutation burden (TMB) (B) and microsatellite instability (MSI) (C) in pan-cancer. (D) The proportion of patients who receive immunotherapy in the low- and high-NCBP2 subgroup in the IMvigor210 cohort (bladder cancer). (E) Kaplan–Meier curve of low- and high-NCBP2 subgroup in the IMvigor210 cohort. (F) The proportion of patients who receive immunotherapy in the low- and high- NCBP2 subgroup in the GSE91061 cohort (melanoma). (G) Kaplan–Meier curve of low- and high- NCBP2 subgroup in the GSE91061 cohort. $*p < 0.05$, $**p < 0.01$, $***p < 0.001$.

We found that NCBP2 was positively correlated with most of the immunomodulatory genes in LIHC, PAAD, PRAD, and UVM, while negatively correlated with most of immunomodulatory genes in CESC, COAD and LUSC (Fig. 6A). To dissect the role of NCBP2 in predicting the efficacy of ICB therapy, we analyzed the correlation between the expression of NCBP2 and TMB/MSI, which showed that NCBP2 was positively correlated with TMB in LUAD and THYM, and a negative association was found in COAD and UCEC (Fig. 6B). For MSI, a positive association in HNSC, KIRC, LUSC, PRAD and THYM, as well as a negative association in COAD and TGCT, was identified (Fig. 6C). Then, we analyzed the predictive role of NCBP2 in patients who received ICB therapy. In the urinary system tumors of the IMvigor210 cohort (bladder cancer), relationship between NCBP2 and the response to anti-PD-L1 treatment was analyzed. The patients exhibited four degrees of response, ranging from progressive disease (PD), stable disease (SD), partial response (PR), to complete response (CR). Results showed that patients with high NCBP2 expression had an anti-PD-L1 response rate of 30.48% (32/105), compared with that of 18.65% (36/193) in patients with low NCBP2 expression (Fig. 6D). Kaplan–Meier survival analysis revealed that patients with high NCBP2 expression had longer OS than those with low NCBP2 expression, and the hazard ratio for univariate cox was 0.62 (0.46–0.85) (Fig. 6E). In addition, the opposite results were found in patients with melanoma who received anti-PD-1 therapy in GSE91061 cohort (Figs. 6F & 6G).

## Tumor stemness and chemotherapeutic value of NCBP2

Cancer stemness can be associated with immunotherapy resistance, but direct clinical evidence is lacking (*Bayik & Lathia, 2021*). We next employed several established algorithms to assess stemness of TCGA samples based on transcriptomic and epigenetic data. These stemness-indexes (mRNAsi, EREG-mRNAsi, mDNAsi, EREG-mDNAsi, ENHsi, DMPsi) derived from the above algorithms represent the degree of resemblance between tumor cells and stem cells. In general, NCBP2 was positively correlated with stemness-indexes in most cancers, except for ACC, CHOL, LAML, PCPG, and SARC (Fig. 7A). Notably, an apparent correlation between NCBP2 expression and all six stemness-indexes was found in LUSC and TCGT (Fig. 7A). These data suggested that NCBP2 might promote cancer progression by regulating cancer stemness in multiple cancers. Subsequently, drug sensitivity analysis was performed to appraised the correlation between NCBP2 and response to drugs in multiple cancer cell lines using data downloaded from the GDSC database, which indicated that NCBP2 was positively associated with sensitivity of 15 drugs (drug resistant), while negatively correlated with sensitivity of 54 drugs (drug sensitive) (Fig. 7B upper part). Furthermore, targets of the above drugs were analyzed, which showed that these NCBP2-associated drugs mainly targeted the DNA replication, chromatin histone methylation, ABL signaling, cell cycle and PI3K signaling (Fig. 7B lower part).

## Functional analysis of NCBP2 at single cell level

As a means of deciphering the role of NCBP2 in cancers at the single-cell levels, CancerSEA was used to investigate its functional states, which showed that NCBP2 was positively associated with cell cycle, DNA damage, DNA repair, invasion and stemness for majority

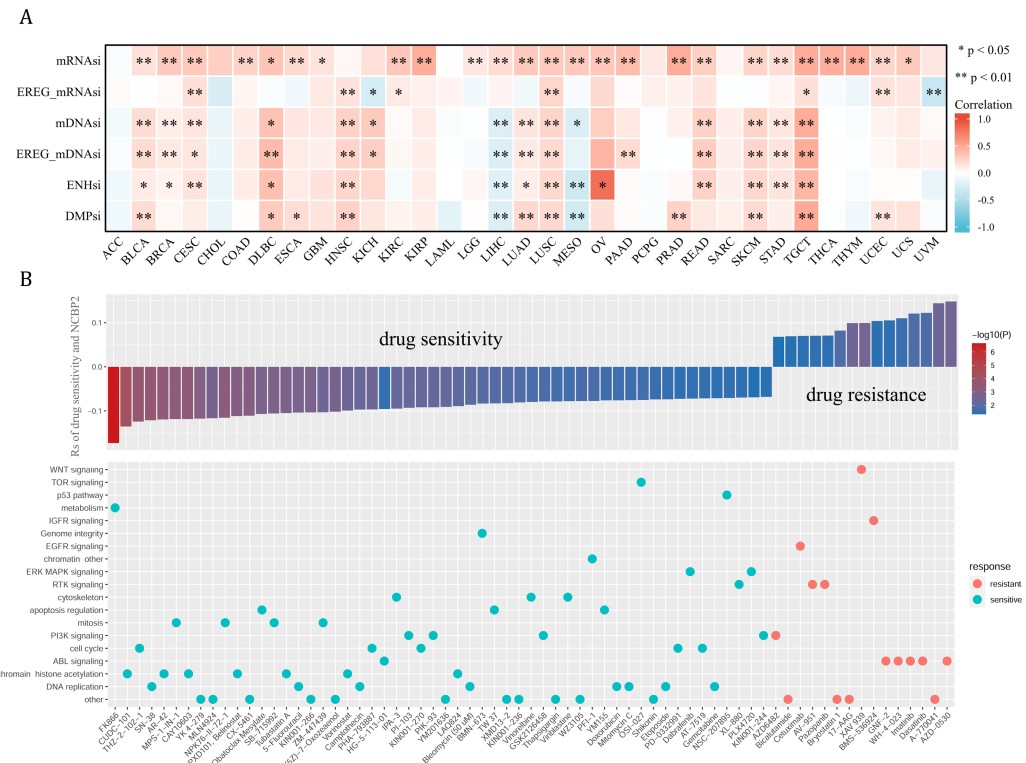

**Figure 7 Tumor stemness and chemotherapeutic value of NCBP2.** (A) Heatmap exhibited the relationship between NCBP2 and stemness indices across cancers. Positive correlation in red and negative correlation in blue. (B) (upper) The relationship between NCBP2 and drug sensitivity calculated by Spearman algorithm. The color of each column represents the *p* value, whereas the height of each column represents the correlation coefficient. Rs represents the drug sensitivity correlated with the NCBP2 expression. (lower) Dot plot visualized the signal pathways targeted by drugs which were sensitivity (blue) or resistant (red) to the NCBP2. The signal pathways were ranked by the frequency of being targeted. *p < 1.05; **p < 0.01.

cancers excluding RB, UM, while negatively associated with apoptosis, inflammation, and hypoxia in majority cancers (Fig. 8A). Subsequently, we further examined the association between NCBP2 and several biological process of specific cancer types. Results indicated that NCBP2 was positive correlated with differentiation, metastasis, proliferation, inflammation and quiescence in AML; negative correlation with DNA repair, DNA damage, apoptosis, metastasis, invasion and quiescence in UM, noting that NCBP2 was positive correlation with angiogenesis, differentiation and inflammation, while negative correlation with DNA repair and cell cycle in RB (Figs. 8B–8D). Moreover, T-SNE diagrams were used to visualize NCBP2 expression profiles at the single-cell level in AML, UM and RB (Figs. 8E–8G).

## DISCUSSION

NCBP2 is a component of the nuclear cap-binding protein complex (CBC), which binds to the monomethylated 5′ cap of nascent pre-mRNA in the nucleoplasm. NCBP2 increased apoptosis mediated cellular organization and brain morphology disruption, aggravating

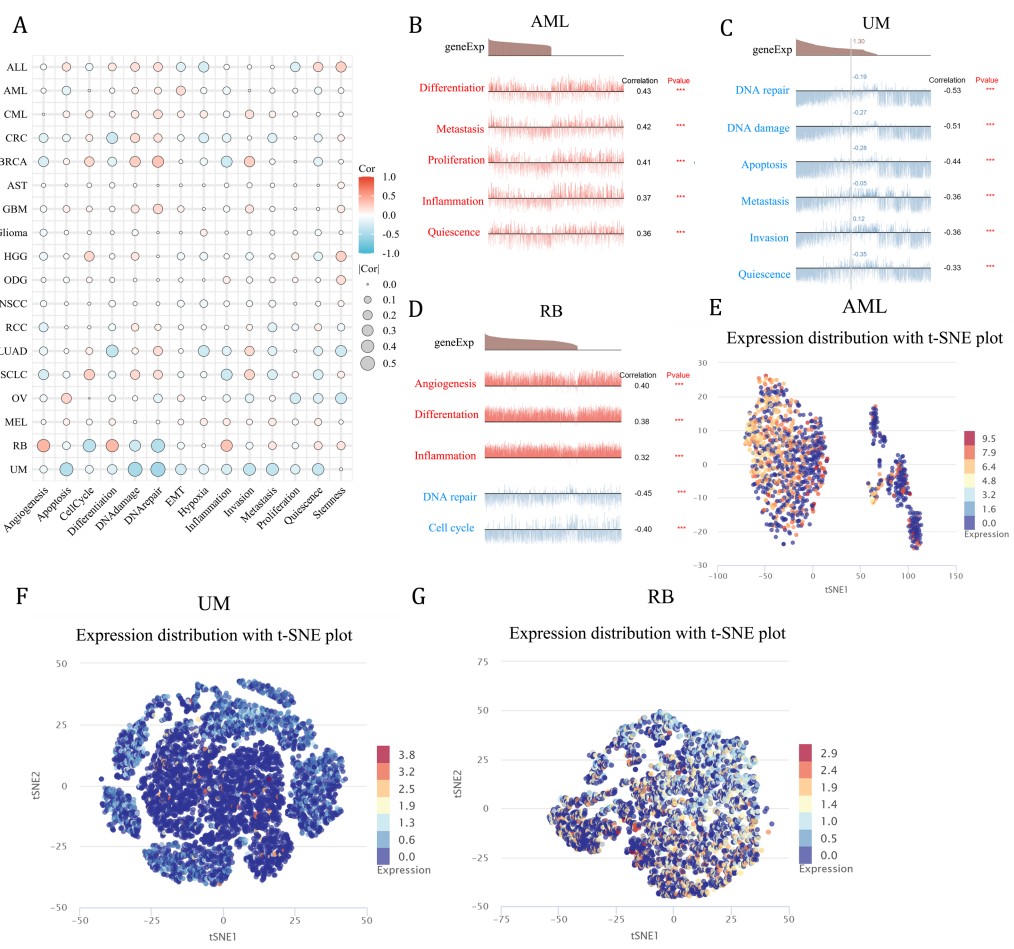

**Figure 8 Functional analysis of NCBP2 at single cell level.** (A) Correlation analysis between NCBP2 and cancer-related functional states at single cell level according to CancerSEA dataset. (B–D) The association between NCBP2 expression and several biological process in AML (B), UM (C), RB (D). (E–G) t-SNE diagrams were used to visualize NCBP2 expression profiles in AML (E), UM (F), RB (G).

neurodevelopmental defects in model organisms (*Barsh et al., 2020*). NCBP2 was elevated in head and neck squamous cell carcinoma tissues (*Xu et al., 2023*), and overexpression of NCBP2 promotes hepatocellular carcinoma cells proliferation and migration (*Zhou et al., 2022*). Moreover, NCBP2 facilitates the translation of c-JUN, a protein that activates the MEK/ERK signaling pathway, thereby promoting the growth and proliferation of pancreatic cancer cells (*Xie et al., 2023*). These studies suggested the oncogenic role of NCBP2 in several specific cancers, however the complicated mechanism by which NCBP2 contributes to cancer progression remain poorly understood. In this study, we identified that a range of cancers exhibit alterations in *NCBP2*, with its expression being up-regulated in the majority of these cancers. Kaplan–Meier and univariate or multivariate Cox regression analysis indicated that NCBP2 serve as a prognostic factor in several cancers, particularly in PAAD and UCEC. Notably, the roles of NCBP2 were complicated and multifaceted across different cancers.

According to the results of GSEA in pan-cancer, NCBP2 was closely related to immune related pathways, such as IFN-alpha response, IFN-gamma response, IL-6/JAK/STAT3 signaling, IL-2/STAT5 signaling, allograft-rejection pathways, inflammatory response, and TNF alpha signaling-via NFKB. Previous studies have shown that IFN-alpha response, IFN-gamma response, IL-6/JAK/STAT3 signaling, IL-2/STAT5 signaling, allograft-rejection pathways, inflammatory response, and TNF alpha signaling were related to clinical response and prognosis of immunotherapy (*Ayers et al., 2017*; *Cheng et al., 2021*; *Diakos et al., 2014*; *Lin et al., 2020*). This information suggested that NCBP2 may be involved in the immune response related pathways of cancers. Recent studies also emphasize the significance of NCBP2 in cancer immunity in various cancers, such as hepatocellular carcinoma and head and neck squamous cell carcinoma (*Xu et al., 2023*; *Zhou et al., 2022*). Therefore, we further explored the role of NCBP2 in pan-cancer immunity. Anti-tumor immune response largely depends on the functional status and composition of tumor immune infiltrates (*Knutson & Disis, 2005*; *Principe et al., 2020*; *Wang et al., 2020*). These immune infiltrates modulate effector cells *via* inhibitory or stimulating signals. Blocking inhibitory or promoting stimulating immune signals can enhance antitumor response. For example, disrupting PD-1/PD-L1 signal, a negative regulatory mechanism for T effector cells, is one of the most popular anti-tumor immunotherapies (*Mahoney, Freeman & McDermott, 2015*). Combining targeted therapy and immunotherapy is a promising strategy for killing cancer cells (*Gotwals et al., 2017*). There has been a great deal of excitement about immunotherapy as a new era of cancer treatment, but its low response rate has limited its application (*Zhang & Zhang, 2020*). However, little was known about the relationship between NCBP2 and immunotherapy. Multiple algorithms (ESTIMATE score, Evaluating the Proportion of Immune and Cancer cells (EPIC), Cell-type Identification By Estimating Relative Subsets Of RNA Transcripts (CIBERSORT), Tumor IMmune Estimation Resource (TIMER), quantitative analysis of immune cell infiltration using RNA sequencing (QUANTISEQ), Microenvironment Cell Populations-counter (MPCOUNTER)) were employed to dissect the immune infiltrates composition of tumor microenvironment, which indicated that NCBP2 was correlated with various immune infiltrates and the expression of multiple immunomodulatory genes. These data suggested that NCBP2 may be interacted with tumor microenvironment composition. High TMB status and MSI were considered prognostic biomarker for ICB (*Chan et al., 2019*; *Yamamoto & Imai, 2019*). Correlation analysis showed a significant correlation between NCBP2 and TMB, as well as MSI in several cancers, such as THYM and COAD. This finding implied that NCBP2 may have prognostic value regarding ICB response. Subsequent analysis demonstrated that elevated NCBP2 expression was linked to improved survival rates and response to anti-PD-L1 therapy in bladder cancer, whereas it was associated with poorer survival outcomes in melanoma. These results suggest that NCBP2 is correlated with the efficacy of immunotherapy and may serve as a potential biomarker for predicting survival and therapeutic response in patients undergoing ICB treatment.

Inspired by the results of NCBP2 and cancer immunity, we speculated that NCBP2 might play a role in cancer stemness. Our findings showed that NCBP2 was positively correlated with stemness indexes in many cancers, suggesting it could be a target to

overcome therapeutic resistance. We discovered that NCBP2 was positively linked to resistance in 15 drugs and negatively linked to sensitivity in 54 drugs, mainly affecting DNA replication, chromatin histone methylation, ABL signaling, and the cell cycle. Thus, patients with high NCBP2 expression might benefit from drugs targeting these resistance-related pathways (*Charaf et al., 2016*; *Lawrence, Daujat & Schneider, 2016*; *Milanovic et al., 2017*; *Song, Wang & Liu, 2022*). Furthermore, we conducted a comprehensive investigation into the significant correlation between NCBP2 and several critical biological processes, including differentiation, metastasis, proliferation, inflammation, quiescence, DNA repair, DNA damage, apoptosis, invasion, angiogenesis, and the cell cycle at the single-cell level. However, the underlying mechanisms driving these associations remain elusive and require further exploration.

In this study, NCBP2 was specifically examined across cancer types, which highlights the potential of NCBP2 from prognostic to immunotherapeutic aspects. However, there are also some limitations. Most of our data come from open-access databases, so clinical cohort validation is needed in the future study. In addition, the mechanism underlying NCBP2 and the tumor microenvironment need further experimental validation. In conclusion, we comprehensively analyzed the role of NCBP2 across cancers, showing its potential in predicting prognosis and immunotherapeutic response. The expression of NCBP2 was correlated with genomic alteration, prognosis, immune response related pathways, tumor microenvironment, immune cell infiltration, TMB, and MSI in various cancers. Our study demonstrates that NCBP2 may serve as an important biomarker for prognosis and immunotherapy response in clinical practice.

## ACKNOWLEDGEMENTS

We acknowledge cBioPortal, UCEC Xena, GEO, TIMER2.0, Human Protein Atlas, STRING, CCLE, GSDC, CancerSEA database and *etc.* for providing their platforms and contributors for uploading their meaningful datasets.

### Funding

This work was sponsored by Natural Science Foundation of Xinjiang Uygur Autonomous Region (2023D01C92). The funders had no role in study design, data collection and analysis, decision to publish, or preparation of the manuscript.

### Grant Disclosures

The following grant information was disclosed by the authors:
Natural Science Foundation of Xinjiang Uygur Autonomous Region: 2023D01C92.

### Competing Interests

The authors declare there are no competing interests.

## Author Contributions

- Shichao Li conceived and designed the experiments, performed the experiments, analyzed the data, prepared figures and/or tables, authored or reviewed drafts of the article, and approved the final draft.
- Yulan Wang conceived and designed the experiments, analyzed the data, prepared figures and/or tables, authored or reviewed drafts of the article, and approved the final draft.
- Xi Yang conceived and designed the experiments, performed the experiments, analyzed the data, prepared figures and/or tables, authored or reviewed drafts of the article, and approved the final draft.
- Miao Li conceived and designed the experiments, performed the experiments, analyzed the data, prepared figures and/or tables, authored or reviewed drafts of the article, and approved the final draft.
- Guoxiang Li conceived and designed the experiments, analyzed the data, prepared figures and/or tables, and approved the final draft.
- Qiangqiang Song conceived and designed the experiments, analyzed the data, prepared figures and/or tables, and approved the final draft.
- Junyu Liu conceived and designed the experiments, performed the experiments, prepared figures and/or tables, authored or reviewed drafts of the article, and approved the final draft.

## Human Ethics

The following information was supplied relating to ethical approvals (i.e., approving body and any reference numbers):

The Medical Ethics Committee of the General Hospital of the Xinjiang Military Command (Urumqi, China) granted approval for the utilization of clinical excisions.

## Data Availability

The datasets generated and/or analyzed are available at:

cBioPortal database (NCBP1, NCBP2, NCBP3, NCBP2-AS1, NCBP2-AS2, NCBP2L): https://www.cbioportal.org/results/oncoprint?cancer_study_list=laml_tcga%2Cacc_tcga%2Cblca_tcga%2Clgg_tcga%2Cbrca_tcga%2Ccesc_tcga%2Cchol_tcga%2Ccoadread_tcga%2Cesca_tcga%2Cgbm_tcga%2Chnsc_tcga%2Ckich_tcga%2Ckirc_tcga%2Ckirp_tcga%2Clihc_tcga%2Cluad_tcga%2Clusc_tcga%2Cdlbc_tcga%2Cmeso_tcga%2Cov_tcga%2Cpaad_tcga%2Cpcpg_tcga%2Cprad_tcga%2Csarc_tcga%2Cskcm_tcga%2Cstad_tcga%2Ctgct_tcga%2Cthym_tcga%2Cthca_tcga%2Cucs_tcga%2Cucec_tcga%2Cuvm_tcga&Z_SCORE_THRESHOLD=2.0&RPPA_SCORE_THRESHOLD=2.0&profileFilter=mutations%2Cgistic&case_set_id=all&gene_list=NCBP1%250ANCBP2%250ANCBP3%250ANCBP2-AS1%250ANCBP2AS2%250ANCBP2L&geneset_list=%20&tab_index=tab_visualize&Action=Submit

UCSC Xena database (TCGA, legacy; https://xenabrowser.net): li, shichao (2025). UCSC expression profile of TCGA. figshare. Dataset. https://doi.org/10.6084/m9.figshare.28399442.v1

XianTao Academic (NCBP2, pan-cancer, prognosis, immunity): https://www.xiantaozi.com/products/apply

MSigDB Database (immune): li, shichao (2025). Msigdb. figshare. Dataset. https://doi.org/10.6084/m9.figshare.28399436.v1

IMvigor210: http://research-pub.gene.com

GSE91061: https://www.ncbi.nlm.nih.gov/geo/query/acc.cgi?acc=GSE91061

Human Protein Atlas, NCBP2: https://www.proteinatlas.org/ENSG00000114503-NCBP2/subcellular

CCLE database (modified) and GDSC database (modified): li, shichao (2025). CCLE&GDSC.rar. figshare. Dataset. https://doi.org/10.6084/m9.figshare.28399427.v1

CancerSEA database: The CancerSEA data was available for peer review but became inaccessible before publication.

TIMER2.0 database (http://timer.cistrome.org/) is a data analysis tool. We entered target gene symbol and the website automatically generated figures.

## Supplemental Information

Supplemental information for this article can be found online at http://dx.doi.org/10.7717/peerj.19050#supplemental-information.

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
