# Peer review of "NCBP2 predicts the prognosis and the immunotherapy response of cancers: a pan-cancer analysis"

_PeerJ, doi:10.7717/peerj.19050_

## Round 0.1 · original submission · Major Revisions

Please address concerns of the reviewers and amend manuscript accordingly.

Reviewer 1 ·

Basic reporting

NA

Experimental design

NA

Validity of the findings

NA

Additional comments

The study explores the expression, genetic alterations, and clinical relevance of Nuclear Cap Binding Protein subunit 2 (NCBP2) across various cancers using multiple cancer databases such as The Cancer Genome Atlas (TCGA) and Genotype-Tissue Expression (GTEx). The research highlights that high expression of NCBP2 is correlated with poor prognosis, enhanced cancer stemness, and diverse immune responses across multiple cancer types. It also finds that the expression of NCBP2 correlated with overall survival rate. The research concludes that NCBP2 might be a novel therapeutic target for immunotherapy.

I have a few comments:
1. How were the data from different sources normalized or adjusted for batch effects, especially given the use of multiple databases like TCGA and GTEx? Authors should state these clearly in methods sections.
2.While the association between NCBP2 expression and cancer characteristics is well-established, the paper could discuss into the mechanistic pathways through which NCBP2 affects these processes. What are some possible mechanisms?
3. There are minor grammatical errors that do not impede understanding but authors should correct them or use professional editing services.

Reviewer 2 ·

Basic reporting

no comment

Experimental design

no comment

Validity of the findings

no comment

Additional comments

Manuscript by Li et.al., highlights the importance of NCBP2 across various cancers, in predicting the prognostic and immunotherapeutic response in clinical practice with the use of open-access datasets.
The manuscript details the extensive computational analysis, revealing that NCBP2 is most frequently altered and differentially expressed among several cancers, playing an important role in the immune response and progression, indicating a potential role in cancer stemness and chemotherapy value.
Overall, the manuscript is well written with clear rationale, experimental design, validation and presentation of the data. Findings and limitations of the studies are appropriately reported.
Thus, I would like to recommend the manuscript for publication. With following minor textual changes
1. The resolutions of Figure 1A and Figure 2A a need to be fixed.
2. The figure legends should be revised with bit more details to understand the figures independent of the manuscript text.
3. The abbreviations need to be expanded when used first in the manuscript, irrespective of the listing in S1.

·

Basic reporting

Introduction/background is poor. See additional comments given below.

Experimental design

Some clarifications needed. See additional comments given below.

Validity of the findings

Conclusions/claims are undue and misleading, and thus need to be revised throughout. See additional comments given below.

Additional comments

The research paper by Li et al. (106114-v0, NCBP2, a novel prognosis and immunity related biomarker for cancers: A pan-cancer analysis).

The authors present data analysis/results on the pan-cancer expression profiles of NCBP2 and show its use in prognosis. They also validate the expression using immunohistochemistry in pa-tient/cancer samples. The writing and overall presentation are good. However, conclu-sions/claims are undue and misleading.



Specific/major comments:

1. Title: “novel prognosis … biomarker” – NCBP2 as a “novel” prognostic biomarker in cancers was already known (for example, PMID:36635327). It is no longer novel.

2. Lines 58-59: “we found that NCBP2 was an oncogene in most cancers” – this is a misleading conclusion. A mere overexpression does not imply causation. It could as well be an effect.

3. Lines 73-76: How studies on eIF4E-dependent translation (ET) emphasize the importance of cap-binding proteins? Revise the mismatch.

4. Lines 77-79: “clinical implications of CBC in cancers are still unclear” – on what basis the authors made such a statement? Why do authors speculate that CBC should have any implica-tions in cancer at all? It appears that the lacunae/problem statement is ill defined. Much of the introduction, especially paragraph 3, is totally irrelevant. Further, authors have not given any background literature for line 91 “Considering the role of cap-binding proteins in cancers” but made a totally speculative statement in line 79 “clinical implications of CBC in cancers are still unclear”. There are numerous recent papers that show the role/clinical implications of NCBP2 in various cancers (for example, PMID:36635327, PMID:38001714, etc.).

5. Lines 92-93: “CBC genomic alteration pattern of pan-cancer was revealed … which had the highest alteration frequency” – give relevant context and citations.

6. Lines 158-160: “bottom 30% as low-NCBP2” – is very ambiguous. Given that NCBP2 is sig-nificantly overexpressed in cancer, is this “bottom 30%” still significantly higher than nor-mal/control? If yes/no, what is the DEG profile of “bottom 30%” in comparison to nor-mal/control?

7. Line 166: How was “normalized enrichment score” computed?

8. Lines 454-455: “expression of NCBP2 was correlated with genomic alteration” – I could not find any such correlation/result in the paper. Fig. 1 does not present such a correlation. Did the authors show that NCBP2 expression trend follows the extent of alterations?

9. Fig. 1 has numerous issues. Fig. 1A is poorly presented/explained and incomprehensible. It is unclear what the mutation/CNA data are, and how they are informative when all are +.

10. Figures should be improved. Fonts should be precise and legible. Many figures/parts are poorly explained. Legends should be self-explanatory and should contain a brief inference of the figure/result.

Reviewer 4 ·

Basic reporting

1. The introduction section is poorly formatted and doesn't reflect the objectives of the manuscript.
2. There is not enough connection between the method and result sections. There is no clear progression of how the authors got those results, for example, the origin of samples and the method of their analysis. Mentioning those methods before presenting results would help readers follow the story.
3. Results are repeated in the discussion section, which makes them redundant.

Experimental design

The authors' objectives in analyzing NCBP2 and its role as an oncogene and potential biomarker of cancer are clearly defined.
1. As there are multiple NCBPs, I suggest using another NCBP (perhaps as a control), which would help highlight NCBP2 as an oncogene and potential therapeutic target.
2. Figure 2B looks out of context and does not add much value to the research story.

Validity of the findings

1. The study's conclusion is not clearly stated as it presents mixed results regarding whether it has a significant role in cancer hallmarks.

---

## Round 0.2 · accepted · Accept

All issues pointed by the reviewers were adequately addressed and amended manuscript is acceptable now.

Reviewer 1 ·

Basic reporting

NA

Experimental design

NA

Validity of the findings

NA

Additional comments

The author has addressed all my concerns.

·

Basic reporting

-

Experimental design

-

Validity of the findings

-

Additional comments

The authors have responded to the reviewer’s comments and revised the manuscript accordingly.